# The Role of MicroRNAs in Proteostasis Decline and Protein Aggregation during Brain and Skeletal Muscle Aging

**DOI:** 10.3390/ijms23063232

**Published:** 2022-03-17

**Authors:** Stephany Francisco, Vera Martinho, Margarida Ferreira, Andreia Reis, Gabriela Moura, Ana Raquel Soares, Manuel A. S. Santos

**Affiliations:** 1Institute of Biomedicine—iBiMED, Department of Medical Sciences, University of Aveiro, 3810-193 Aveiro, Portugal; s.marquesfrancisco@ua.pt (S.F.); veramartinho@ua.pt (V.M.); margaridamccferreira@ua.pt (M.F.); areis@ua.pt (A.R.); gmoura@ua.pt (G.M.); 2Multidisciplinary Institute of Aging, MIA-Portugal, Faculty of Medicine, University of Coimbra, Rua Largo 2, 3º, 3000-370 Coimbra, Portugal

**Keywords:** miRNA, mammalian tissue aging, age-related protein aggregation, proteostasis network

## Abstract

Aging can be defined as the progressive deterioration of cellular, tissue, and organismal function over time. Alterations in protein homeostasis, also known as proteostasis, are a hallmark of aging that lead to proteome imbalances and protein aggregation, phenomena that also occur in age-related diseases. Among the various proteostasis regulators, microRNAs (miRNAs) have been reported to play important roles in the post-transcriptional control of genes involved in maintaining proteostasis during the lifespan in several organismal tissues. In this review, we consolidate recently published reports that demonstrate how miRNAs regulate fundamental proteostasis-related processes relevant to tissue aging, with emphasis on the two most studied tissues, brain tissue and skeletal muscle. We also explore an emerging perspective on the role of miRNA regulatory networks in age-related protein aggregation, a known hallmark of aging and age-related diseases, to elucidate potential miRNA candidates for anti-aging diagnostic and therapeutic targets.

## 1. Introduction

By 2050, 22% of the world’s population will be over 60 years of age, which will lead to an increase in the incidence of age-associated diseases and impact the longevity and well-being of the elderly [1]. Alterations in proteostasis are characteristic of age-associated neurodegenerative diseases like Alzheimer’s and Parkinson’s, and also of normal aging [2,3,4]. Proteostasis decline consists of gradual imbalances in protein synthesis, folding, and degradation processes over time that lead to increased protein misfolding and proteotoxic stress that ultimately reduce organismal longevity [2,5,6,7,8].

Production of misfolded proteins results in protein aggregation and in the activation of proteostasis-related stress response pathways, which are collectively known as the proteostasis network (PN) and involve the heat shock response (HSR), the unfolded protein response in the endoplasmic reticulum (ER) (UPR^ER^) and in the mitochondria (UPR^mt^), the ubiquitin–proteasome system (UPS), and the autophagy–lysosomal pathway (ALP). The activation of the PN is necessary to refold and/or degrade protein aggregates, which diminishes stress levels and prevents cell death [9,10,11,12]. Protein aggregation during normal aging has also been studied in several organismal models, namely in the kidney and pancreas of *Rattus norvegicus* [13,14], in the heart, bone marrow, and spleen of *Mus musculus* [15,16], and in the human cortex [17,18]. However, the underlying mechanisms of this phenomenon remain unclear to date.

MicroRNAs (miRNAs) are a class of small, non-coding RNAs 21 to 23 nucleotides in length that regulate gene expression post-transcriptionally by targeting and preventing the translation of specific messenger RNAs (mRNA). They regulate a plethora of cellular processes, namely cell growth and differentiation, organismal development, physiological functioning, and cell homeostasis [19,20,21]. In fact, the ~2500 miRNAs reported in humans regulate roughly 60% of protein-coding genes, while each miRNA regulates, on average, 200 target genes through complementary target sites [22]. Over the past decade, miRNA overexpression and/or knockdown experiments have been shown to directly alter lifespan in *C. elegans*, *D. melanogaster*, and *M. musculus* [23,24,25,26]. Notably, studies in centenarians have provided more knowledge about how dynamic human miRNA profiles are throughout aging [27,28] For instance, increased biogenesis of miRNAs is observed in centenarians compared with octogenarians [29]. More recent studies showed that miRNAs regulate age-associated processes, and some evidence links miRNAs to protein aggregation in mammalian tissues, namely in the brain and skeletal muscle [30,31]. 

Nevertheless, miRNA expression can be tissue-specific or cell-specific, and at the same time, miRNAs can be non-cell autonomous, acting as mediators across various tissues [32]. Age-related alterations that affect specific miRNA families may differ based on cell and tissue type, making these analyses challenging due to the complexity and interconnectedness of the miRNA networks in a single organism [19,26]. In addition, circulating miRNAs (c-miRNAs) and their shuttles (i.e., extracellular vesicles) were recently noted as markers of healthy physiological aging, with miR-19a-3p and miR-19b-3p found to be increased in aging individuals but decreased in healthy centenarians, thus adding another layer of intricacy to the miRNA regulatory network [27,28]. For this reason, past studies have provided a compilation of miRNAs associated with specific aging hallmarks in various model organisms to provide a better representation of the miRNAs that are linked to aging across several tissues [33]. This review focuses on recently published reports on the miRNA regulation of gene targets involved in the proteostasis-related processes that are relevant to tissue aging, with particular emphasis on mammalian tissue aging. 

## 2. The Biogenesis and Target Binding of miRNAs

RNA polymerase II transcribes miRNAs, producing primary miRNA transcripts (pri-miRNAs) that are then processed by Drosha/DGC8, an RNase III enzyme microprocessor complex, to produce ~70–100 nucleotide (nt) stem–loop precursor miRNAs (pre-miRNAs) [21,34]. These pre-miRNAs are exported from the nucleus into the cytoplasm via Exportin-5, a RanGTP-dependent, double-stranded RNA-binding and nuclear-transport protein, and then cleaved by the RNase III enzyme Dicer, yielding 22-nt double-stranded mature miRNA duplexes [35,36]. The miRNA may originate from the 5′ side of the pre-miRNA, called the “5p” strand, or from the 3′ end, which is known as the “3p” strand [37]. In the past, the process of strand selection was reported to begin with one of the duplexed miRNA strands known as the miR or guide strand, which was selected and loaded into the Argonaute protein to form the miRNA-induced silencing complex (miRISC), while the other strand, traditionally denoted as the passenger strand or miR*, was ejected from the complex and degraded [37]. Nevertheless, in some cases, either strand can be selected and loaded into the Argonaute proteins to regulate gene expression, as is the case of miR-34 where miR-34b-5p and miR-34b-3p are present in approximately equal concentrations in humans, allowing for the targeting of distinct mRNAs [37,38,39]. The phenomenon referred to as “miRNA arm switching”, is where the preferential selection of either 3′ or 5′ miRNA arms for strand selection can be dependent either on tissue-type or on the developmental stage of life, while the de-regulation of this process is associated with disease [37,38,39]. The miRISC can target specific mRNAs via sequence complementarity between the seed sequence of the loaded miRNA guide strand (2–7 nucleotides) and in most cases, the 3′ untranslated regions (3′UTR) of the target mRNAs [32,34]. In some cases, the 3′ ends of miRNAs can undergo preferential complementary binding to motifs in the 5′UTR regions of the target mRNAs [40,41]. Arm switching can result in the accumulation of mature miRNAs with altered seed sequences, which drastically alters the targeting of mRNAs [42]. During aging, there is a general shift in the expression of the 3′ and 5′ mature arms of miRNAs, with increasing 5′ mature expression and decreasing 3′ expression over time, especially for miR-6786 in human plasma samples [42]. At the same time, miR-4423 was identified with the most decreased 5′ expression ratio, namely in breast milk, the heart, testis, stem cells, and blood cells, thus signifying that age-related variations in 3′ to 5′ mature miRNA expression ratios may be tissue-dependent [42].

In the age of next-generation sequencing, it was shown that one single miRNA locus is able to produce several distinct miRNA isoforms or isomiRs, which are miRNA sequence variants that have altered 3′ and/or 5′ end(s) due to nucleotide addition, deletion, or substitution [43]. The biogenesis of isomiRs can be template dependent, resulting in 5′ or 3′ nucleotide shifting, or non-template dependent, leading to post-transcriptional RNA editing and tailing [44]. IsomiRs can be classified as 3′, 5′, polymorphic, and mixed, depending on the sequence variation and alterations in length [44]. Post-maturation sequence modifications such as trimming (removal of nucleotides via exoribonucleases at the 3′ end) and tailing (addition of nucleotides by terminal nucleotidyl transferases at the 3′ end) give rise to 3′ isomiRs (also known as 3′-isomiR-trimmed and 3′-isomiR-tailed) [43,44,45]. It also has been reported that 5′ isomiRs are largely generated by the imprecise cleavage of miRNA sequences via Drosha and/or Dicer [43]. Polymorphic isomiRs contain changes within the mature sequence and are unchanged in length, while mixed isomiRs contain alterations in length and sequence [46,47]. Notably, the generation of isomiRs was shown to be cell- and tissue-specific in mammals, and they were used as biomarkers of cancer and implicated in Alzheimer´s disease [48,49].

There are very few studies that fully elucidate the role of isomiRs in the context of aging. Recently, isomiR-19a-3p was found to be significantly increased in unhealthy centenarians when compared with healthy centenarians, indicating that this isomiR can be used as a potential marker for healthy aging [28]. Furthermore, isomiR sequencing of metformin-treated (lifespan-extension treatment) human umbilical-vein endothelial cells (HUVECs) during replicative senescence revealed that non-canonical sequences accounted for almost 40% of the total miRNA pool, whereby metformin treatment significantly altered the relative abundance of 133 isomiRs which were identified as variants of 73 individual miRNAs [50]. Interestingly, target genes of these miRNAs and isomiRs were found to be part of the phosphatidylinositol-3-kinase (PI3K)–Akt pathway and the mammalian target of the rapamycin (mTOR) signaling pathway [50]. Notably, most isomiRs affected by metformin treatment were identified as 3′ isomiRs; however, around 5% were identified as 5′ isomiRs, which were shown to shift the seed sequence and yielded a greater number of target genes that were not common in the miRNA targets in this study [50]. Once again, isomiRs appear to be effective markers for evaluating healthy aging and lifespan extension. Nevertheless, the biological repercussions of the dynamic changes in isomiRs through aging and senescence should be further explored.

Studies reported that at least a six-base-pair match of miRNA to mRNA is necessary for gene-expression silencing, and due to this complementary binding with the target mRNA, individual miRNAs can have multiple target mRNAs (approximately 100 mRNAs) and can simultaneously target different 3′-UTR sites of the same mRNA, while different miRNAs can also target the same mRNA, which augments the complexity of the regulatory outcomes for the same mRNA [32]. This signifies that each cell and tissue type present very complex patterns of miRNAs, and the dysregulation of one or more components within the miRNA networks can result in homeostasis imbalance, shortened lifespan, and disease [51,52]. In fact, centenarians display upregulation of the expression of miRNAs, namely miR-16, miR-18a, miR-21, while the mRNA expression of RNA Polymerase II, Drosha, Exportin-5, and Dicer are also all upregulated compared with octogenarians [29].

## 3. Target-Directed miRNA Degradation and Its Role in Fundamental Cellular Processes

In contrast to miRNA biogenesis, the role of miRNA decay/degradation—especially of target-directed miRNA degradation (TDMD)—in health and disease is only beginning to be elucidated. TDMD occurs when a target RNA is able to induce the degradation/decay of its cognate miRNA instead of eliciting target repression, which allows for the modulation of a variety of cellular processes. TDMD occurs when there is extensive complementary binding at the 3′ region of the miRNA, which prompts a conformational change that stimulates the 3′ end to be released from the binding pocket of the Ago PAZ domain [53,54,55]. Subsequent exposure of the 3′ end allows for enzymatic degradation of the miRNA [54]. This process may also be accompanied by post-transcriptional modifications of the miRNA sequence, such as tailing (by terminal nucleotidyl transferases) and trimming (by 3′-to-5′ exonucleases), resulting in the production of isomiRs [56,57]. TDMD targets are also able to trap Ago2 in a particular conformation that allows for the exposure of the miRNA 3′ end for tailing and trimming while continually bound to Ago2 [54,55]. Interestingly, it is possible to measure miRNA decay rates in mammalian cells by combined methodologies such as metabolic RNA 4sU-based “pulse-chase” labelling with high-throughput RNA sequencing, wherein trimming and tailing were shown to be dynamic over time in fibroblasts [57]. These methodologies may prove to be useful in future studies evaluating miRNA decay throughout physiological aging.

A handful of studies have explored endogenous TDMD and its function in mammals, identifying the key target genes involved in this phenomenon. For instance, Serpine1, which encodes the serine-threonine protease inhibitor, acts as a TDMD target to control miR-30b-5p and miR-30c-5p degradation in mouse fibroblasts, thus promoting the cell-cycle re-entry of quiescent fibroblasts [58]. Interestingly, all miR-30 family members (miR-30a, -30b, -30c, -30d, and -30e) are able to interact with Serpine1; however, only miR-30b and miR-30c exhibit an extended 3′ end complementarity (also known as 3C pairing) to the Serpine1 transcript, proving that this 3C pairing is fundamental to TDMD [58]. Taken together, the Serpine1:miR-30b/c interaction plays a key regulatory role in mammalian cell phenotypes by facilitating cell-cycle re-entry of quiescent cells. Another example of an endogenous TDMD target, the neuronal regeneration-related protein (Nrep) gene, was reported to direct miR-29b degradation through 3′ trimming, controlling overall motor coordination and motor learning in mice [59]. Normally, Nrep restricts miR-29b expression solely to cerebellar Purkinje neurons, while the disruption of the miR-29 site in Nrep expanded miR-29b expression to the cerebellar granular layer, leading to abnormal motor learning and coordination in mice [59]. More specifically, the scrambling of the miR-29 site in Nrep in neural progenitor cells resulted in the absence of miR-29 isoforms produced via 3′ trimming or tailing [59]. Interestingly, the Nrep miR-29 site was also shown to have high sequence similarity with the non-coding RNA libra in zebrafish, which regulates anxiety-like and explorative behavior in zebrafish [59]. Taken together, TDMD, through endogenous targets, is essential in maintaining proper mammalian brain functions and behaviors and in controlling cellular phenotypes.

## 4. Expression Profiles of miRNA Are Dynamic and Are Tissue-Specific throughout Aging

During tissue aging, there is a progressive decline in miRNA abundance, which was first reported due to the age-associated decreases in miRNA biogenesis that occur through the downregulation of Dicer [60]. Indeed, miRNA biogenesis was shown to improve stress tolerance and longevity in C. elegans and in adipose tissue in mice, while specific miRNAs were associated with stress resistance and aging [23,60]. More recently, it was found that miRNA-biogenesis genes are, in fact, highly targeted by the miRNAs that are involved in age-related processes, namely miRNA-71, which was shown to downregulate global miRNA expression and lead to increased mRNA expression variability with age [19]. These results are in line with previous evidence that established miR-71 as one of the most studied miRNAs in the context of aging and longevity, with its expression being significantly upregulated throughout early-to-middle adulthood [24,25]. Interestingly, miR-71 was also recently reported to stimulate ubiquitin-dependent protein turnover, specifically in the intestine, lengthening the lifespan of *C. elegans*, while its inhibition affected organismal proteostasis [61]. In particular, food odors sensed by *C. elegans* via ciliated AWC olfactory neurons 1) stimulate a cell-nonautonomous regulation of ubiquitin-dependent protein degradation via the ubiquitin–proteasome system (UPS) and via endoplasmic-reticulum (ER)-associated protein degradation (ERAD) and 2) augment heat stress resistance in the intestine [61]. The authors showed that miR-71 specifically inhibits Toll-receptor-domain protein (TIR-1) in amphid wing “C” (AWC) olfactory neurons, while miR-71 and/or TIR-1-knockout worms display UPS and ERAD dysfunction, eliminating the influence of the food source in this response [61]. One important point that arises is that miRNA regulatory networks are extensively intricate in several organismal models, and miRNA expression profiles appear to be both tissue-specific and, at the same time, interconnected with various tissues.

Most studies that have explored the role of miRNAs in normal mammalian tissue aging have focused on experiments using samples mainly from brain and skeletal muscle tissue, perhaps due to these tissues’ propensity for the development of age-related diseases [62]. One highly studied miRNA in aging and age-related diseases is miR-34. This miRNA was shown to be upregulated in aging *C. elegans* [63], while its overexpression extends lifespan in Drosophila and reduces the propensity for age-related neurodegeneration [64]. However, contrary to other organisms, members of the miR-34 family play a detrimental role in mammalian brain aging, where upregulation of miR-34c was found in the mouse hippocampus in both normal aging and AD models and is associated with cognitive decline [65]. Moreover, age-associated upregulation of miR-29a and miR-29b in the mouse brain leads to a dysregulation of the microglia and an increase in neuroinflammation, also a hallmark of brain aging [66]. Similarly, during the aging process, muscle degeneration and regeneration are imbalanced, leading to a loss in muscle homeostasis. Additionally, miRNAs, namely miR-29, were linked to sarcopenia, an age-related loss of muscle function, and were shown to modulate apoptosis, senescence, and insulin-like growth factor (IGF-1) signaling in aging muscle cells [67]. Thus, miRNAs can be used as biomarkers of aging in more than one tissue and for specific age-associated phenotypes such as sarcopenia.

Recently, a group of studies explored the role of miR-206 during vascular (namely cardiac muscle) and skeletal muscle aging and homeostasis [68,69,70,71,72,73,74]. Augmented expression of miR-206 was linked to vascular aging, especially in individuals with arrhythmias, [74] and was also shown to inhibit the proliferation of vascular smooth muscle cells and to promote atherosclerosis [72,73]. These studies emphasize the importance of exploring miRNA regulation in one specific mammalian tissue in order to be able to find new therapeutic targets for specific, age-related pathologies.

Taking all of these recent findings into consideration, it is evident that miRNA regulatory networks mediate aging and longevity to some extent. However, it is essential to clarify the roles of miRNAs for each organismal model due to the complex nature of miRNA regulatory networks and the plethora of potential targets that each miRNA may possess. In the following sections, we highlight more closely the impact of miRNAs’ specifically age-related proteostasis decline that involves protein degradation and clearance processes, summarized in Figure 1, and delve into their potential influence on widespread protein aggregation during mammalian aging.

## 5. The Role of miRNA Regulation in Age-Related Autophagic Decline in Brain and Skeletal Muscle

Described as major modulators of several proteostasis-related degradation pathways during mammalian tissue aging, scientific reports have focused on miRNA regulatory networks involving the autophagy–lysosome pathway (ALP), with particular emphasis on the mammalian target of the rapamycin (mTOR) complex, mainly in the brain and muscle [24,75]. Inukai and colleagues (2012) conducted one of the first studies using Solexa deep sequencing to identify the link between the differential expression of miRNAs during brain aging and mTOR signaling [24]. In this study, most of the differentially expressed miRNAs declined in relative abundance in 24–25-month-old mice compared with 5-month-olds, with KEGG enrichment analysis revealing that miRNAs, namely miR-5620 isomiR and miR-341 isomiR, are involved in mTOR/ protein kinase B (Akt)/Forkhead box class O (FOXO) signaling [24].

One type of autophagy, called macrophagy, involves the degradation and recycling of the cytoplasmic components, including cell debris and misfolded proteins, that are packed into double-membrane vesicles called autophagosomes, which are fused with lysosomes to be degraded by lysosomal hydrolases [75]. Protein aggregates can be engulfed by autophagosomes and degraded, and aging may alter autophagic clearance through defects in autophagosome formation, failed autophagosome–lysosome fusion, and lysosome acidification [76]. Age-related autophagic dysfunction may also result in the development of age-related diseases, namely sarcopenia and neurodegenerative diseases. For an extensive review, see [75]. During skeletal muscle aging, autophagic activity was shown to decrease in elderly patients with sarcopenia and in murine muscle tissue, where the protein levels of LC3, a marker of autophagosomes and autolysosomes, and E1-like enzyme ATG7 decline with age [62]. It was recently confirmed that miR-34a is a key player in autophagic activity in brain aging, with its upregulation resulting in defective autophagy and abnormal mitochondrial dynamics in d-galactose-induced rat models of aging [77]. In this same study, autophagic dysfunction was reversed via the administration of a miR-34a inhibitor to d-galactose-induced SH-SY5Y cells and resulted in the upregulated expression of autophagy-related proteins, namely LC3, Beclin 1, ATG7, and the degradation of P62 [77]. Taken together, miRNA-34a may be an effective therapeutic target for attenuating and/or reversing age-related autophagic decline in the mammalian brain.

Muscle-enriched miRNAs, known as myo-miRNAs, such as miR-1 and miR-206, can regulate targets through the PI3K/AKT/mTOR/FOXO pathway, which controls cell-cycle, proliferation, and differentiation processes in myocytes [68,69,70,71]. Recently, hemodialysis patients undergoing regular physical resistance training were shown to have lower levels of miR-206, which resulted in greater myogenesis and fewer cardiac calcifications, a hallmark of vascular aging [69]. The authors concluded that, after regular training, downregulation of miR-206 possibly stimulates myogenesis through the insulin-like growth factor 1 (IGF1) binding and activation of the PI3K/AKT/mTOR pathway [70]. Similarly, in human skeletal muscle, 26 miRNAs were identified to be regulated by age, exercise, or a combination of both, whereby nine of these miRNAs, namely miR-99a-5p, miR-99b-5p, miR-100-5p, miR-199a, and miR-196b-5p, have validated target sequences within the 3′ UTRs of the target genes involved in the Akt/mTOR-signaling pathway, such as mTOR, Akt, regulatory associated protein of MTOR complex 1 (RPTOR), and IGF1 [71].

Notably, miR-378 was also identified to be crucial to muscle homeostasis by sustaining autophagy through direct targeting of phosphoinositide-dependent protein kinase 1 (PDK1), promoting Akt-mTORC1 signaling, and reducing myocyte apoptosis via caspase 9 targeting [69]. MiR-378-knockout mice exhibited impaired autophagy and the accumulation of defective mitochondria, while miR-378 overexpression was shown to reduce phosphorylation of unc-51-like autophagy activating kinase 1 (ULK1), subsequently promoting the formation of autophagosomes, a result which was validated by upregulated LC3 expression and puncta in the C2C12 myotube cell line [69]. This study is important because it confirms that miR-378 serves a beneficial role in promoting autophagy and identifies its two direct targets: (1) PDK1, which activates Akt and mTORC1 to promote autophagy in skeletal muscle and (2) Caspase 9, which suppresses myocyte apoptosis. Nevertheless, there are still many aspects of the miRNA regulation of autophagy to be explored in the context of tissue aging, especially in mammalian species.

In the context of age-related protein aggregation, miR-1 was shown to regulate muscle function and improve pharyngeal pumping in response to proteotoxic stress over time, while also suppressing polyglutamine 35 (polyQ35) protein aggregation in *C. elegans* [30]. A v-ATPase subunit, vha-13 was identified as a direct target of miR-1, which regulates lysosomal biogenesis and function [30]. Thus, miR-1 should be studied as a putative therapeutic strategy for combating muscle-specific protein aggregation and improving muscle motility across the lifespan. Previously, increased levels of miR-1 were shown to promote autophagy, thus reducing the accumulation of protein aggregates in *C. elegans* and mammalian cells [78]. Notably, in mouse cortical neurons and HeLa cells, miR-1 activates autophagy through the targeting of Tre-2/Bub2/CDC16(TBC) Rab GTPase-activating protein (TBC1D15) in response to the accumulation of mutant huntingtin aggregates [78]. TBC1D15 was shown to block autophagy by inactivating Rab7, which regulates autophagosome and lysosome fusion. Once again, miR-1 appears to be an effective candidate for anti-aging therapeutics that target autophagy-related processes.

## 6. miRNAs Mediate the Ubiquitin-Proteosome System in Brain Aging and Muscle Atrophy

The ubiquitin–proteasome system (UPS) is a cellular proteolytic degradation process consisting of the 26S proteasome as its central component, encompassing two 19S regulatory components and one 20S core [79]. Proteasomal activity decreases during normal aging and in neurodegeneration [80]. Within the last year, two studies have described the contribution of miRNA regulation on UPS-related alterations throughout aging in the non-disease context [81,82,83]. For example, miR-127-5p was found to have reduced expression in the brains of aged C57BL/6J mice undergoing LPS-induced ischemia, and for the first time, identified 26S proteasome non-ATPase regulatory subunit 3 (Psmd3) as one of its targets [82]. This study is important because the authors not only provided a brain miRNome through mature miRNA sequencing but also identified a novel miRNA in the brain, miR-127, which is associated with cell apoptosis and proteasomal activity during aging and ischemia. Furthermore, in healthy carriers of the apolipoprotein E (ApoE ε4) polymorphism, linked to Aβ clearance and heightened risk for AD development, miR-153-3p was shown to be increased, leading to blockage of nuclear factor erythroid-2-related-factor-2 (Nrf2)-mediated proteasomal function and erythrocyte Aβ accumulation in plasma [81]. In addition, proteasome-associated components, such as plasma Kelch-like ECH-associated protein 1 (Keap1) and erythrocyte histone deacetylase 6 (HDAC6), were also increased in Apoeε4 carriers in comparison with non-carriers [81]. The ApoE ε4 polymorphism, along with the proteoasome-associated components and miR-153-3p, can be used as plasma or circulating markers for evaluating vulnerability to Aβ accumulation and associated proteasomal dysfunction during brain aging, and, at the same time, for evaluating the propensity for the development of pathological age-related neurodegeneration and protein aggregation, thus providing a non-invasive measure for carriers.

Muscle atrophy is characterized by aberrant protein degradation through the UPS pathway [83]. In skeletal muscle, muscle ring finger 1 (MuRF1) and muscle atrophy F-box (MAFbx)/atrogin-1 are E3 ubiquitin ligases that are thought to be involved in the ubiquitination of substrates for degradation by the 26S proteasome, wherein increases in these UPS components promote muscle atrophy [83]. During muscle atrophy, there is overexpression of MuRF1 and MAFbx/atrogin-1, while inhibition of these components slows muscle loss [84,85]. Several studies have highlighted the role of miRNAs in the regulation of MuRF1 and MAFbx/atrogin-1 expression. For instance, miR-23a was shown to inhibit the translational activation of MuRF1 and MAFbx/atrogin-1, counteracting muscle atrophy in a dexamethasone-induced muscle atrophy mouse model [86]. In addition, increased expression of muscle-specific miR-1 was found to reduce HSP70 levels, decreasing AKT phosphorylation and leading to activation of FOXO3 and ultimately resulting in the upregulation of MuRF1 and MAFbx/atrogin-1 expression during dexamethasone-induced muscle atrophy in mice [87]. In particular, upon dexamethasone induction, miR-1, regulates the dephosphorylation and subsequent activation of FoxO3a through HSP70/Akt signaling, which directly or indirectly inhibits proteins that counteract muscle atrophy [87]. More recently, inhibition of MuRF1 and MAFbx/atrogin-1 transcription was shown to improve muscle loss in the skeletal muscle (specifically, the gastrocnemius muscles) of 40-wk-old senescence-accelerated mice prone 8 (SAMP8) mice when compared with age-matched senescence-accelerated mouse resistant (SAMR1) mice following stimulation with epigallocatechin-3-gallate (EGCG), a catechin component in green tea known to ameliorate muscle loss [88]. EGCG was shown to inhibit myostatin and to upregulate miR-486-5p, which directly suppresses phosphatase and tensin homolog (PTEN), augmenting Akt phosphorylation and resulting in the inhibition of active FoxO1a and the suppression of MuRF1 and MAFbx/atrogin-1 transcription in the skeletal muscle of 40-wk-old SAMP8 mice as well as in late passage C2C12 myoblast cells [88]. This study uncovered a unique muscle-loss intervention using the EGCC component of green tea to alter miRNA regulation in the ubiquitin–proteasome process and PI3K/Akt/FOXO signaling in skeletal muscle. In fact, these findings provide evidence that miRNA-associated UPS activation promotes muscle atrophy primarily through the targeting of MuRF1 and MAFbx/atrogin-1 in skeletal muscle.

## 7. miRNAs as UPR and Protein-Aggregation Regulators

Under stress conditions, misfolded proteins accumulate in the ER lumen and elicit the UPRER for the degradation of these proteins via three key pathways, namely the protein kinase RNA-like ER kinase (PERK), the inositol-requiring enzyme-1 alpha (IRE-1α)/X-box binding protein-1 (XBP-1), and the activating transcription factor 6 (ATF6) pathways [89,90]. UPR^ER^ activity is reduced during aging ER stress conditions, while overexpression of XBP-1 is able to enhance ER stress resistance and longevity [89,90]. Overall, it is evident that the UPR^ER^ plays a functional role in aging and lifespan.

miRNAs were reported to either directly modulate the ER stress response or, in contrast, undergo regulation via ER stress [91]. In fact, ER stress sensors, namely PERK, can directly regulate miRNA expression coordinating pro- and anti-apoptotic signaling upon ER stress [92]. For instance, miR-30c-2-3p via PERK-mediated signaling leads to a decrease in XBP-1 mRNA, which subsequently promotes cell death in NIH-3T3 fibroblasts [92]. During ER stress, the downregulation of the miR-106b-25 family via PERK activation allows for the activation of pro-apoptotic B-cell lymphoma 2 (BCL-2) family genes such as Bim, thus eliciting ER-stress-induced apoptosis in cell lines [93]. The downregulation of the miR106b-25 family and ER-stress-induced apoptosis was also found in an ALS mouse model (mutant SOD1 G93A), which links this miRNA family and its targets to neurodegeneration in ALS [93]. Recently, in the cortical and hippocampal neurons of mice, upon stimulation with advanced glycation of the end products of bovine serum albumin (AGE-BSA) to induce ER stress, miR-24, -27b, -124, -224, -290, -351, and -488 were found to be downregulated, while the mRNA of UPR targets (Perk, Ire1α, Chop, and Puma) were upregulated, thus indicating that these miRNAs are probable UPR regulators [94]. Using miRNA-target-prediction algorithms, the authors found that PERK was targeted by miR-24 and miR-488, Ire1α by miR-24, -124, -290, -351, and -488, Chop by miR-224, and Puma by miR-24, -27b, and -351 [94]. Upon exposure to sugars, the advanced glycation of proteins results in interactions between advanced glycation end products (AGEs) and RAGE (receptor of AGEs), which promotes the accumulation of aberrant toxic proteins, subsequent cellular oxidative stress, and ER stress [94]. In addition, some of these miRNAs were also found in expression-profiling studies in neurodegeneration, namely miR-27b, -124, and -488 [95], suggesting that these miRNAs can be suitable markers of aging in both contexts. Nevertheless, a direct causal relationship between protein aggregation and miRNA dysregulation remains to be established in the context of both healthy aging and age-related pathological neurodegeneration.

## 8. miRNA-Binding Proteins also Regulate Lifespan through Proteostasis Network Components

Numerous reports have demonstrated that miRNA-binding proteins regulate aging through proteostasis network components. In *C. elegans*, for example, Argonaute, Dicer, Drosha, and DGCR8/Pasha were shown to control aging and longevity [19,24,25]. Argonaute-1 (alg-1) was shown to promote longevity, whereas Argonaute-2 (alg-2) shortens lifespan where many of the differentially expressed genes in mutant alg-1 and alg-2 C. elegans are part of the insulin/IGF-1 signaling (ILS) pathway, with regulation via DAF-16/ FOXO [96]. Furthermore, specifically in alg-1 mutants, heat shock factor 1 (hsf1), a transcriptional factor that is part of the heat shock stress response, was found to be downregulated [96]. Huntingtin aggregation cell cultures and mouse models as well as post-mortem samples from patients with Huntington’s disease have also shown that Argonaute-2 (AGO2), a core component of RISC, re-localizes and accumulates in the stress granules located in the striatal neurons that express mutant Huntingtin (mHTT) aggregates, thus hindering late autophagosome and lysosome fusion and, at the same time, leading to a global increase in miRNA expression [31]. However, due to the re-localization of AGO2 and the formation of AGO2–miRNA complexes in stress granules, the authors noted that overall miRNA activity, especially target silencing, was hindered in mHTT-expressing neurons [31]. In this case, protein aggregates were shown to interfere with autophagic clearance, which promotes AGO2 accumulation and alterations in miRNA levels and activity, possibly leading to neuronal damage. These results present a novel perspective, summarized in Figure 2, which highlights the role of age-related protein aggregation in autophagic dysfunction, miRNA biogenesis, and miRNA activity during mammalian aging (Figure 2).

miRNA biogenesis was associated with stress tolerance and lifespan extension in mouse tissues, whereby Dicer activity was shown to decrease over time in adipose, skeletal muscle, and brain tissue [24,97,98]. Mori and colleagues (2012) found 136 miRNAs were downregulated in aged (24-month-old) C57BL/6 mice, accounting for approximately 51% of all differentially expressed miRNAs in adipose tissue [60]. In addition, Dicer protein levels decreased significantly in aged mice and in human pre-adipocytes, which was reversed by calorie restriction, while Dicer overexpression in the intestine of C. elegans upregulated heat shock genes hsp-70 and hsp-70-like (F44E5.4) [60]. More recently, fat-specific Dicer-knockout (Adicer KO) mice exhibited accelerated insulin resistance and shortened lifespans and showed increased mTORC1 activation in both adipose tissue and skeletal muscle [97]. Furthermore, an age-associated decline in Dicer was also reported to contribute to the deregulated mature miRNA expression profiles in dopaminergic neurons in the ventral midbrain of old C57Bl/6 N mice, whereby 78% of them were found to be downregulated compared with young mice [98]. Furthermore, in the same study, pharmacological stimulation of Dicer activity with enoxacin was shown to facilitate dopamine neuron survival by reducing vulnerability specifically to thapsigargin-induced ER stress, suggesting that miRNA biogenesis via Dicer is neuroprotective against ER stress. Downregulation of Dicer during aging may compromise miRNA biogenesis and lead to accumulated ER stress and the neurodegeneration of dopamine neurons [98]. Innovative sequencing techniques such as photoactivatable ribonucleoside-enhanced crosslinking and immunoprecipitation (PAR-CLIP) sequencing have identified alternative functions of Dicer in human cells as well as in *C. elegans*, whereby Dicer was shown to be involved in the degradation of several structural RNAs that were previously undetectable with other methodologies [99].

Moreover, Drosha can cleave hairpin structures that are embedded in mRNAs, destabilizing them and therefore directly controlling mRNA expression. Drosha was shown to cleave mRNAs for differentiation factors that are fundamental for neurogenesis [100]. In addition, upon viral infection, Drosha was reported to be exported into the cytoplasm to cleave viral genomic RNA in order to suppress viral replication [101,102]. Phosphorylation and nuclear export of Drosha also occurs following heat shock and oxidative stress, impairing Drosha-mediated miRNA biogenesis [103]. Another innovative sequencing technique, called formaldehyde crosslinking immunoprecipitation and sequencing (fCLIP-seq), has uncovered that Drosha cleaves non-canonical substrates originating from non-miRNA loci into short hairpins, generating small RNAs [104] Protocols using PAR-CLIP and fCLIP can identify potential interactions between miRNA-binding proteins and structural RNAs in tissue samples and in the context of age-related diseases [104]. Taken together, the results of these studies provide new research avenues for the targeting of miRNA biogenesis components to change miRNA expression, and possibly novel mRNA targets, in order to augment proteostasis network efficiency in clearing toxic proteins and attenuate age-related protein aggregation.

## 9. Conclusions

The main aim of this review was to provide supportive evidence, based on recent studies, for the exciting new prospect that miRNAs, and to some extent miRNA-binding proteins, can directly regulate proteostasis, which is relevant to organismal lifespan expansion, and they can also be used as markers of healthy tissue aging for comparisons with age-related diseases. Notably, miRNA target genes in proteostasis-related degradation pathways such as the UPS and ALP systems appear to be highly tissue-specific in mammalian brain and skeletal muscle during aging, across various organismal models. In addition, the role of miRNA regulation in the age-related dysfunction of protein degradation and clearance mechanisms has become even more evident in recent studies of mammalian brain and skeletal muscle tissues. Based on the findings that demonstrate miRNA-mediated proteostasis regulation during mammalian aging, we theorize that future studies should explore a direct causal relationship between protein aggregation and miRNA dysregulation in the context of both healthy aging and age-related pathological neurodegeneration and muscle atrophy, which are represented in Figure 2. Since there are some overlapping miRNA profiles altered in both the normal and pathological contexts of aging, the potential to modulate miRNAs, especially augmenting levels of miRNAs that target protein clearance and autophagy-related genes, could be relevant for future anti-aging therapeutic strategies.

## Figures and Tables

**Figure 1 ijms-23-03232-f001:**
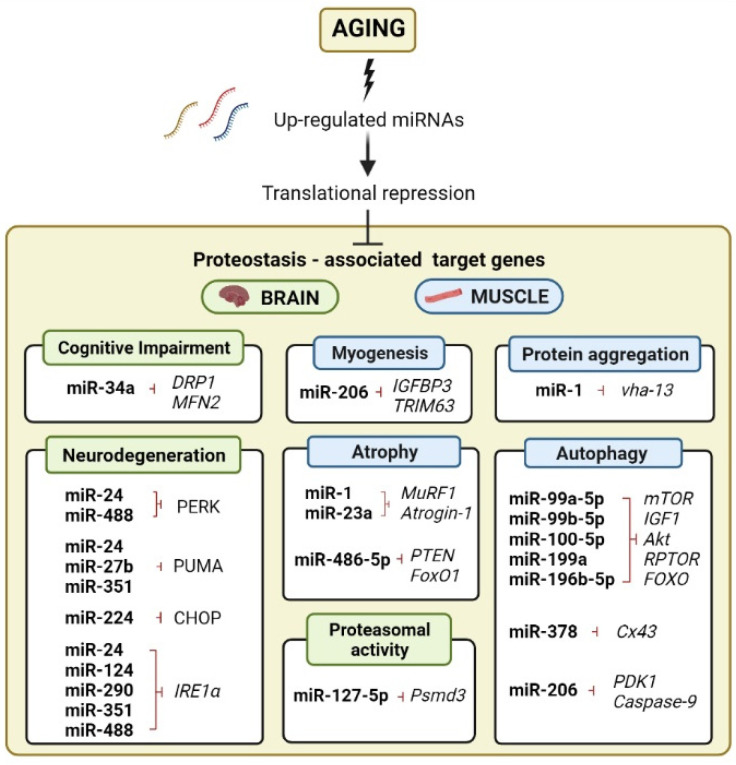
Role of miRNAs in proteostasis imbalance during mammalian aging. Aging can lead to the upregulation of miRNAs in brain and skeletal muscle tissue, repressing the translation of proteostasis-associated target genes. MiR-34a, miR-24, miR-27b, miR-124, miR-224, miR-290, miR-351, miR-488, and miR-127-5p have proteostasis-associated target genes, mainly pertaining to the unfolded protein response (UPR) and the ubiquitin–proteasome system (UPS), which are implicated in mammalian brain aging (green headings). MiR-1, miR-23a, miR-99a-5p, miR-99b-5p, miR-100-5p, miR-199a, miR196b-5p, miR-206, miR-378, and miR-486-5p modulate target genes that are part of the autophagy–lysosomal pathway (ALP), with many genes belonging to the AKT/mTOR/FOXO signaling pathways. These miRNAs have been linked specifically to skeletal-muscle aging in mammals in the non-disease context (blue headings).

**Figure 2 ijms-23-03232-f002:**
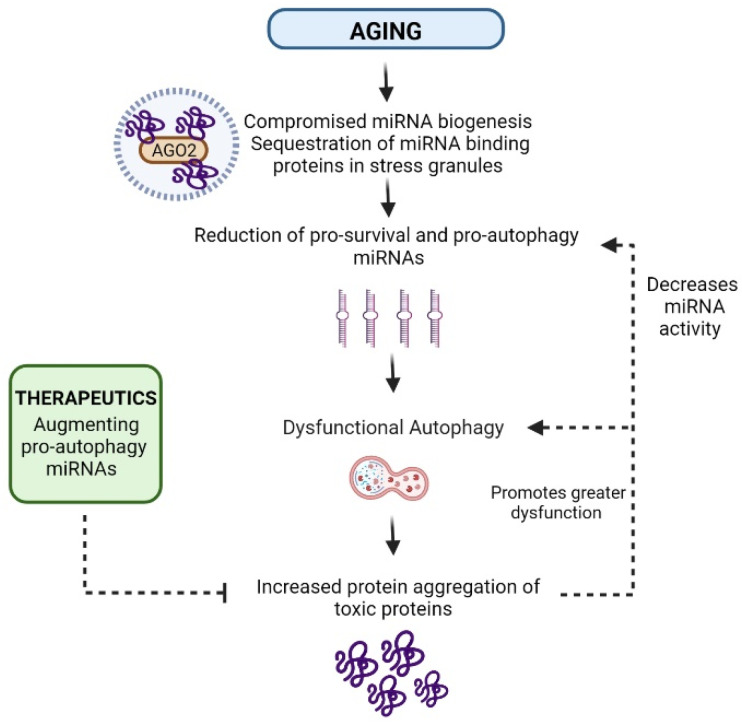
The interplay of aging and age-related protein aggregation in miRNA expression and proteostasis imbalance in mammalian tissues. During aging, alterations in miRNA biogenesis lead to an overall reduction or imbalance in pro-survival and pro-autophagy miRNA levels. Due to these changes, autophagic degradation becomes increasingly hindered over time, thus leading to increased accumulation of toxic proteins. With the rise of this toxic protein aggregation, autophagic clearance mechanisms become oversaturated and inefficient in degrading toxic proteins. At the same time, it is believed that the presence of toxic protein aggregation hinders miRNA activity, with miRNA binding proteins such as Argonaute-2 (AGO2) and AGO2–miRNA complexes being sequestered in stress granules and hindering their modulation of protein degradation target genes such as autophagy-associated genes. One effective intervention strategy involves augmenting the levels of pro-autophagy miRNAs in order to deal with the accumulation of toxic protein aggregates over time.

## Data Availability

Not applicable.

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
