# Peer review of "The Role of MicroRNAs in Proteostasis Decline and Protein Aggregation during Brain and Skeletal Muscle Aging"

_ijms, 2022, doi:10.3390/ijms23063232_

Round 1
Reviewer 1 Report
The role of microRNAs on proteostasis decline and protein aggregation during brain and skeletal muscle aging
Author(s): Stephany Francisco, Vera Martinho, Margarida Ferreira, Andreia Reis, Gabriela Moura, Ana Raquel Soares, Manuel A. S. Santos
Manuscript type: Review
This review by Francisco et al describes the role of microRNAs in protein aggregation and proteostasis in the context of aging. Remarkably, the authors present an extensive review of the current literature in vast areas such as miRNA biogenesis, multiple mechanisms of pathogenesis and individual examples. Given the complexity and the vast literature available, this is an excellent review.
Please, note my suggestions to improve the coverage of the literature below:
Major Comments:
- The miRISC can target specific mRNAs via sequence complementarity between the seed sequence of the loaded miRNA guide strand (2-7 nucleotides) and the 5´untranstranslated region (5´UTR) of the target mRNAs [29].” Although there are examples of miRNAs that bind to the 5’UTR, the vast majority of reported targets bind to the 3’UTR.
- “... with 195 KEGG enrichment analysis revealing that miRNAs namely miR-5620 isomiR, and miR- 196 341 isomiR”. The concept of isomiRs and isomiR biogenesis should be introduced and discussed (PMID: 30953728).
- In addition to the arm switch, it might be worth discussing the potential role of isomiRs during aging and anti-aging strategies (PMID: 34160893, PMID: 33311640).
- The paper describes miRNA biogenesis very well, however miRNA homeostasis relies on the equilibrium of biogenesis and decay. Although the latter remains less explored, it is relatively well-established the concept of target-directed miRNA decay (TDMD), which involves a direct degradation of the miRNA rather than transient binding to a binding site. This mechanism is triggered by target RNAs with extensive complementarity between the miRNA and the target, especially on the 3’ end (PMC4388616, PMC6559361, reviewed in PMC6175985), triggering the dislocation of the miRNA from the AGO PAZ domain (PMC6754277, PMC7265490). One well characterized example of TDMD, is the miR-29-binding site located within the 3′ UTR NREP, the disruption of which leads to impaired coordination and motor learning on zebrafish and mice models (PMID: 29483647). I think this should be discussed by the authors in the review.
- I think it should be noted that both Drosha and Dicer have some other roles outside of miRNA biogenesis through other RNA substrates (PMID: 28431232, 25416952) that might be also relevant in aging.
Minor comments:
- I think Figure 2 should describe in more detail the mechanism under “compromised miRNA biogenesis”.
Author Response
Dear Reviewer,
We appreciate the helpful suggestions and commentary on our review entitled, “The role of microRNAs on proteostasis decline and protein aggregation during brain and skeletal muscle aging” for the IJMS Special Issue “RNA Regulatory Networks 2.0”. We would like Reviewer 1 to consider our responses and revisions to the major and minor comments given in the first round of revisions.
In response to the suggestions of Reviewer #1 Major comment #1,
- The miRISC can target specific mRNAs via sequence complementarity between the seed sequence of the loaded miRNA guide strand (2-7 nucleotides) and the 5´untranstranslated region (5´UTR) of the target mRNAs [29].” Although there are examples of miRNAs that bind to the 5’UTR, the vast majority of reported targets bind to the 3’UTR.
We agreed to alter this statement with the following sentences:
“The miRISC can target specific mRNAs via sequence complementarity between the seed sequence of the loaded miRNA guide strand (2-7 nucleotides) and in most cases, the 3´untranstranslated region (3´UTR) of the target mRNAs [29, 33]. In some cases, the 3´ends of miRNAs can have preferential complementary binding to motifs in the 5´UTR region of target mRNAs [39, 40].”
In response to Reviewer #1 Major comments #2 and #3,
- “... with 195 KEGG enrichment analysis revealing that miRNAs namely miR-5620 isomiR, and miR- 196 341 isomiR”. The concept of isomiRs and isomiR biogenesis should be introduced and discussed (PMID: 30953728).
- In addition to the arm switch, it might be worth discussing the potential role of isomiRs during aging and anti-aging strategies (PMID: 34160893, PMID: 33311640).
We decided to describe the concept of isomiRs and biogenesis of isomiRs as well as the potential role during aging and anti-aging strategies in Section 2. miRNAs biogenesis and target binding
“In the age of next generation sequencing, it has been shown that one single miRNA locus is able to produce several distinct miRNA isoforms or isomiRs, which are miRNA sequence variants that have altered 3′- and/or 5′-end(s) due to nucleotide addition, deletion, or substitution respectively [42]. The biogenesis of isomiRs can be template dependent resulting in 5′ or 3′ nucleotide shifting or non-template dependent leading to post-transcriptional RNA editing and tailing [43]. IsomiRs can be classified as 3´, 5´, polymorphic, and mixed depending on the sequence variation and alterations in length [43]. Post-maturation sequence modifications such as trimming (removal of nucleotides via exoribonucleases at the 3´ end) and tailing (addition of nucleotides by terminal nucleotidyl transferases at the 3′ end), give rise to 3’ isomiRs (also known as 3´isomiR-Trimmed and 3’ isomiR-Tailed) [42- 44]. It also has been reported that 5’ iso-miRs are largely generated by imprecise cleavage of miRNA sequences via Drosha and/or Dicer [42]. Polymorphic isomiRs have changes within the mature sequence and are unchanged in length while mixed isomiRs have alterations in length and sequence [45, 46]. Notably, the generation of isomiRs have been shown to be cell- and tissue-specific in mammals and have been used as biomarkers of cancer and implicated in Alzheimer´s disease [47, 48].
There are very few studies that fully elucidate the role of isomiRs in the context of aging. Recently, isomiR-19a-3p was found to be significantly increased in unhealthy centenarians compared to healthy centenarians, indicating that this isomiR can be used as a potential marker for healthy aging [49]. Furthermore, isomiR sequencing of metformin-treated (lifespan extension treatment) human umbilical vein endothelial cells (HUVECs), during replicative senescence revealed that non-canonical sequences accounted for almost 40% of the total miRNA pool whereby metformin treatment significantly altered the relative abundance of 133 isomiRs which were identified as variants of 73 individual miRNAs [50]. Interestingly, target genes of these miRNAs and isomiRs were found to be part of the phosphatidylinositol-3-kinase (PI3K)/Akt and the mammalian target of rapamycin (mTOR) signaling pathway [50]. Notably, most isomiRs affected by metformin treatment were identified as 3´isomiRs however, around 5% were identified as 5´isomiRs which were shown to shift the seed sequence and yielded a greater amount of target genes that were not common to the miRNA targets in this study [50]. Once again, isomiRs appear to be effective markers for evaluating healthy aging and lifespan extension. Nevertheless, the biological repercussions of the dynamic changes of isomiRs through aging and senescence to be further explored.”
In response to Reviewer #1 major comment #4,
- The paper describes miRNA biogenesis very well, however miRNA homeostasis relies on the equilibrium of biogenesis and decay. Although the latter remains less explored, it is relatively well-established the concept of target-directed miRNA decay (TDMD), which involves a direct degradation of the miRNA rather than transient binding to a binding site. This mechanism is triggered by target RNAs with extensive complementarity between the miRNA and the target, especially on the 3’ end (PMC4388616, PMC6559361, reviewed in PMC6175985), triggering the dislocation of the miRNA from the AGO PAZ domain (PMC6754277, PMC7265490). One well characterized example of TDMD, is the miR-29-binding site located within the 3′ UTR NREP, the disruption of which leads to impaired coordination and motor learning on zebrafish and mice models (PMID: 29483647). I think this should be discussed by the authors in the review.
In fact, we decided that the topic TDMD is relevant to our review topic and so, we added a section to our review paper about TDMD and re-numbered the subsequent sections accordingly:
“3. Target-directed miRNA degradation and its role on fundamental cellular processes
In contrast to miRNA biogenesis, the role of miRNA decay/degradation especially target-directed miRNA degradation (TDMD) in health and disease is only beginning to be elucidated. TDMD occurs when a target RNA is able to induce degradation/decay of its cognate miRNA, instead of eliciting target repression, allowing for modulation of a variety of cellular processes. TDMD occurs when there is extensive complementary binding at the 3´region of the miRNA prompting a conformational change that stimulates the 3´end to be released from the binding pocket of the Ago PAZ domain [54–56]. Subsequent exposure of the 3´end allows for enzymatic degradation of the miRNA [55]. This process may also be accompanied by post-transcriptional modifications of the miRNA sequence, such as tailing (by terminal nucleotidyl transferases) and trimming (by 3´-to-5´ exonucleases), resulting in the production of isomiRs [57,58]. TDMD targets are also able to trap Ago2 in a particular conformation that allows for the exposure of the miRNA 3´end for tailing and trimming while continually bound to Ago2 [55,56]. Interestingly, it is possible to measure miRNA decay rates in mammalian cells by combined methodologies such as metabolic RNA 4sU-based “pulse-chase” labelling with high-throughput RNA sequencing wherein trimming and tailing have been shown to be dynamic over time in fibroblasts [58]. These methodologies may prove to be useful in evaluating miRNA decay throughout physiological aging in future studies.
A handful of studies have explored endogenous TDMD and its function in mammals, identifying key target genes involved in this phenomenon. For instance, Serpine1, which encodes serine-threonine protease inhibitor, acts as a TDMD target to control miR-30b-5p and miR-30c-5p degradation in mouse fibroblasts, promoting cell cycle re-entry of quiescent fibroblasts [59]. Interestingly, all miR-30 family members (miR-30a, -30b, -30c, -30d, -30e) are able to interact with Serpine1 however, solely miR-30b and miR-30c have an extended 3′ end complementarity (also known as 3C pairing) to the Serpine1 transcript, proving that this 3C pairing is fundamental to TDMD [59]. Taken together, the Serpine1:miR-30b/c interaction plays a key regulatory role on mammalian cell phenotypes by facilitating cell cycle re-entry of quiescent cells. Another example of an endogenous TDMD target, the neuronal regeneration-related protein (Nrep) gene has been reported to direct miR-29b degradation through 3´trimming, controlling overall motor coordination and motor learning in mice [60]. Normally, Nrep restricts miR-29b expression to solely cerebellar Purkinje neurons while disruption of the Nrep miR-29 site expanded miR-29b expression to the cerebellar granular layer leading to abnormal motor learning and coordination in mice [60]. More specifically, scrambling of the miR-29 site in Nrep in neural progenitor cells resulted in the absence of miR-29 isoforms produced via 3′ trimming or tailing [60]. Interestingly, the Nrep miR-29 site was also shown to have high sequence similarity with the noncoding RNA libra in zebrafish which regulates anxiety-like and explorative behavior in zebrafish [60]. Taken together, TDMD through endogenous targets is essential in maintaining proper mammalian brain functions and behaviors and in controlling cellular phenotypes.”
In response to Reviewer #1 major comment #5:
- I think it should be noted that both Drosha and Dicer have some other roles outside of miRNA biogenesis through other RNA substrates (PMID: 28431232, 25416952) that might be also relevant in aging.
We agree that the roles of Drosha and Dicer outside of miRNA biogenesis may be relevant to aging studies. We included the following paragraph in Section 8. miRNA-binding proteins also regulate lifespan through proteostasis network components
“Innovative sequencing techniques such as photoactivatable ribonucleoside-enhanced crosslinking and immunoprecipitation (PAR-CLIP) sequencing has identified alternative functions of Dicer in human cells as well as in C.elegans, whereby Dicer was shown to be involved in the degradation of several structural RNAs that were previously undetectable with other methodologies [100]. Likewise, Drosha can cleave hairpin structures that are embedded in mRNAs, destabilizing them, and therefore directly controlling mRNA expression. Drosha has been shown to cleave mRNAs for differentiation factors that are fundamental for neurogenesis [101]. In addition, upon viral infection, Drosha has been reported to be exported into the cytoplasm to cleave viral genomic RNA in order to suppress viral replication [102,103]. Phosphorylation and nuclear export of Drosha also occurs following heat shock and oxidative stress, impairing Drosha-mediated miRNA biogenesis [104]. Another innovative sequencing technique called formaldehyde crosslinking immunoprecipitation and sequencing (fCLIP-seq) has uncovered that Drosha cleaves non-canonical substrates originating from non-miRNA loci into short hairpins, generating small RNAs [105] Protocols using PAR-CLIP and fCLIP can identify potential interactions of miRNA-binding proteins and structural RNAs in tissue samples and in the context of age-related diseases [105]. Taken together, the results of these studies render new research avenues for the targeting of miRNA biogenesis components to change miRNA expression and possibly novel mRNA targets, in order to augment proteostasis network efficiency to clear toxic proteins and attenuate age-related protein aggregation.”
In response to Reviewer #1 minor comment:
- I think Figure 2 should describe in more detail the mechanism under “compromised miRNA biogenesis”.
We added to Figure 2 “Sequestration of miRNA binding proteins in stress granules.”
Thank you for taking our revisions into consideration, we look forward to receiving your feedback.
Best regards,
Manuel A.S. Santos and Ana Raquel Soares
Corresponding Authors
Reviewer 2 Report
This is a sound and comprehensive review. I found minor issues that should be addressed by the Authors.
-citation in the text [2]-[4] instead [2-6] or [17],[18] instead [17,18]
-Table 1 and Figure 1 are the same therefore, I suggest choosing one of them.
-Most data regarding C.elegans or mices. It is worth to add paragraph concerning only human
Author Response
Dear Reviewer,
We appreciate the helpful suggestions and commentary on our review entitled, “The role of microRNAs on proteostasis decline and protein aggregation during brain and skeletal muscle aging” for the IJMS Special Issue “RNA Regulatory Networks 2.0”. We would like Reviewer 2 to consider our responses and revisions to comments given in the first round of revisions.
In response to Reviewer #2 minor comment #1:
- citation in the text [2]-[4] instead [2-6] or [17],[18] instead [17,18]
The in-text formatting has been changed accordingly to IJMS Reference Style
In response to Reviewer #2 minor comment #2:
- Table 1 and Figure 1 are the same therefore, I suggest choosing one of them.
We decided to eliminate Table 1 to avoid redundance of the information presented in Figure 1.
In response to Reviewer #2 minor comment #3:
- Most data regarding C.elegans or mices. It is worth to add paragraph concerning only human
Although there are very few studies that report age-related changes in miRNA expression that are tissue-specific in healthy aging and proteostasis, we included the few studies that explore miRNA expression in centenarians in the introduction as well as throughout the text.
“Notably, studies in centenarians have provided more knowledge about how dynamic miRNA profiles are in humans throughout aging [27,28] For instance, increased biogenesis of miRNAs is observed in centenarians compared to octogenarians [29]…
In addition, circulating miRNAs (c-miRNAs) and their shuttles (i.e., extracellular vesicles) have been recently noted as markers of healthy physiological aging, with miR- 19a- 3p and miR- 19b- 3p found to be increased in old individuals but decreased in healthy centenarians, adding another layer of intricacy into the miRNA regulatory network [27,28].”
Thank you for taking our revisions into consideration, we look forward to receiving your feedback.
Best regards,
Manuel A.S. Santos and Ana Raquel Soares
Corresponding Authors